# FrAug: Frequency Domain Augmentation for Time Series Forecasting

## Abstract

Data augmentation (DA) has become a *de facto* solution to expand training data size for deep learning. With the proliferation of deep models for time series analysis, various time series DA techniques are proposed in the literature, e.g., cropping-, warping-, flipping-, and mixup-based methods. However, these augmentation methods mainly apply to time series classification and anomaly detection tasks. In time series forecasting (TSF), we need to model the fine-grained temporal relationship within time series segments to generate accurate forecasting results given data in a look-back window. Existing DA solutions in the time domain would break such a relationship, leading to poor forecasting accuracy. To tackle this problem, this paper proposes simple yet effective frequency domain augmentation techniques that ensure the semantic consistency of augmented data-label pairs in forecasting, named *FrAug*. We conduct extensive experiments on eight widely used benchmarks with several state-of-the-art TSF deep models. Our results show that FrAug can boost the forecasting accuracy of TSF models in most cases. Moreover, we show that FrAug enables models trained with 1% of the original training data to achieve similar performance to the ones trained on full training data, which is particularly attractive for cold-start forecasting. Finally, we show that applying test-time training with FrAug greatly improves forecasting accuracy for time series with significant distribution shifts, which often occurs in real-life TSF applications. Our code is available at `https://anonymous.4open.science/r/Fraug-more-results-1785`.

## 1 Introduction

Deep learning is known for its high demand for training data. Insufficient training data can lead to convergence issues and overfitting in deep models. However, collecting and labeling real-world data can be costly and time-consuming, data augmentation (DA) has become a *de facto* solution to expand the training dataset size for performance improvement (Cubuk et al., 2019).

While various time series DA techniques have been proposed in the literature with the proliferation of deep models for time series analysis, most of them are focused on tasks such as classification and anomaly detection (AD) (Wen et al., 2021).This is partly because the time series forecasting (TSF) task often comes with natural labels without requiring additional labeling efforts. However, the TSF task may also suffer from severe data scarcity problems. However, the TSF task can also suffer from data scarcity, particularly in cold-start forecasting scenarios where little or no historical data is available, such as predicting sales for new products. Additionally, even with sufficient historical data, significant distribution shifts in the forecasting horizon can render the historical data less useful (Pham et al., 2022). Under such circumstances, we need to retrain the forecasting model to fit the new data distribution. Consequently, effective DA techniques are crucial for improving TSF performance.

In supervised learning, data augmentation involves creating artificial data-label pairs from existing labeled data. Ensuring the semantic consistency of these modified data-label pairs is crucial. In the computer vision (CV) field, several previous works have examined the issue of "augment ambiguity," which refers to the introduction of inconsistencies that deteriorate model performance instead of improving it (Gong et al., 2021; Wei et al., 2020). DA for image data is relatively less ambiguous compared to time series data, as images are easier to interpret, allowing for semantics-preserving augmentations such as rotation and cropping/masking irrelevant regions.

In contrast, time series data consist of events generated from complex dynamic systems. We are interested in the temporal relationship among continuous data points in the series. Any perturbations that disrupt this relationship can affect the semantics of the data-label pair, which should reflect the behavior of the underlying system. Thus, care must be taken to ensure that the semantics of the data-label pair is preserved. For time series classification, window cropping (Cui et al., 2016; Le Guennec et al., 2016), window warping (Wen et al., 2021), and noise injection (Wen & Keyes, 2019) can be performed without changing the classification labels, as long as they do not alter the class. Similarly, label expansion techniques (Gao et al., 2020) that manipulate the start and end points of sequence anomalies have shown improvements in the anomaly detection task.

However, time series forecasting is a regression task that requires modeling the fine-grained temporal relationship within a timing window. This window consists of data points in the "look-back window" and the "forecasting horizon," which serve as the data and label for training TSF models, respectively. Aggressive augmentation methods like cropping or warping can introduce missing values or changes in periodicity, which are not suitable for TSF models. In other words, augmented data-label pairs for TSF should be more stringent compared to other time series analysis tasks, an aspect that has not been thoroughly explored in the literature.

In this paper, we propose that well-designed data augmentations in the frequency domain can preserve the fine-grained temporal relationship within a timing window. In dynamic systems that generate time series data, the forecastable behavior is often driven by periodic events[1]. By identifying and manipulating these events in the frequency domain, the generated data-label pairs remain faithful to the underlying dynamic system. This motivation leads us to introduce two simple yet effective frequency domain augmentation methods for TSF, called *FrAug*. Specifically, FrAug performs *frequency masking* and *frequency mixing*, which randomly eliminate some frequency components of a timing window or mix up the same frequency components of different timing windows. Experimental results on eight widely-used TSF benchmark datasets demonstrate that FrAug improves the forecasting accuracy of various deep models, particularly when the size of the training dataset is small, such as in cold-start forecasting or forecasting under distribution shifts.

Specifically, the main contributions of this work include:

- To the best of our knowledge, this is the first work that systematically investigates data augmentation techniques for the TSF task.

- We propose a novel frequency domain augmentation technique named *FrAug*, including two simple yet effective methods (i.e., frequency masking and frequency mixing) that preserve the semantic consistency of augmented data-label pairs in forecasting. In our experiments, we show that FrAug alleviates overfitting problems of state-of-the-art (SOTA) TSF models, thereby improving their forecasting performance.

- FrAug enables models trained with 1% of the original training data to achieve similar performance to the ones trained on full training data in some datasets, which is particularly attractive for cold-start forecasting problems.

- We further design a test-time training policy and apply FrAug to expand the training dataset size. We experimentally show that such a strategy greatly mitigates the distribution shift problem, thereby boosting forecasting accuracy considerably. In particular, for the ILI dataset with severe distribution shifts, we can achieve up to 30% performance improvements.

## 2 RELATED WORK AND MOTIVATION

Data augmentation methods are task-dependent. In this section, we first analyze the challenges in time-series forecasting in Sec. 2.1. Next, in Sec. 2.2, we survey existing DA techniques for time series analysis and discuss why they do not apply to the forecasting task. Finally, we discuss the motivations behind the proposed FrAug solution in Sec. 2.3.

---

[1]The measurements contributed by random events are not predictable.

## 2.1 CHALLENGES IN TIME SERIES FORECASTING

Given the data points in a look-back window $x = \{x_1^t, ..., x_C^t\}_{t=1}^L$ of multivariate time series, where $L$ is the look-back window size, C is the number of variates, and $x_i^t$ is the value of the $i_{th}$ variate at the $t_{th}$ time step. The TSF task is to predict the horizon $\hat{x} = \{\hat{x}_1^t, ..., \hat{x}_C^t\}_{t=L+1}^{L+T}$, the values of all variates at future T time steps. When the forecasting horizon $T$ is large, it is referred to as a long-term forecasting problem, which has attracted lots of attention in recent research (Zhou et al., 2021; Zeng et al., 2022).

Deep learning models have dominated the TSF field in recent years, achieving unparalleled forecasting accuracy for many problems, thanks to their ability to capture complicated temporal relationships of the time series by fitting on a large amount of data. However, in many real applications, time series forecasting remains challenging for two main reasons.

First, the training data can be scarce. Unlike computer vision or natural language processing tasks, time series data are relatively difficult to scale. Generally speaking, for a specific application, we can only collect real data when the application is up and running. Data from other sources are often not useful (even if they are the same kinds of data), as the underlining dynamic systems that generate the time series data could be vastly different. In particular, lack of data is a serious concern in cold-start forecasting and long-term forecasting. In cold-start forecasting, e.g., sales prediction for new products, we have no or very little historical data for model training. In long-term forecasting tasks, the look-back window and forecasting horizon are large, requiring more training data for effective learning. Over-fitting caused by data scarcity is observed in many state-of-the-art models training on several benchmark time-series datasets.

Second, some time-series data exhibits strong distribution shifts over time. For example, promotions of products can increase the sales of a product and new economic policies would significantly affect financial data. When such a distribution shift happens, the model trained on previous historical data cannot work well. To adapt to a new distribution, a common choice is re-training the model on data with the new distribution. However, data under the new distribution is scarce and takes time to collect.

Table 1: Results of different models trained on the ETTh1 dataset augmented by existing augmentation methods. We can observe that these methods will degrade performance in most cases. The metric is MSE, the lower, the better. Detailed implementation is shown in the Appendix.

| Method | Origin | Noise | Freq-Noise | Mask-Rand. | Mask-Seg. | Flipping | Warping |
|---|---|---|---|---|---|---|---|
| DLinear | 0.373 | **0.371** | 0.373 | 0.803 | 0.448 | 0.544 | 0.401 |
| FEDformer | **0.374** | 0.397 | 0.377 | 0.448 | 0.433 | 0.420 | 0.385 |
| Autoformer | **0.449** | 0.476 | 0.467 | 0.608 | 0.568 | 0.446 | 0.465 |
| Informer | 0.931 | 1.112 | 1.044 | **0.846** | 1.013 | 0.955 | 1.265 |

## 2.2 DATA AUGMENTATION METHODS FOR TIME SERIES ANALYSIS

Various time series DA techniques are proposed in the literature. For the time series classification task, many works regard the series as a waveform image and borrow augmentation methods from the CV field, e.g., window cropping (Le Guennec et al., 2016), window flipping (Wen et al., 2021; Semenoglou et al., 2023), and Gaussian noise injection (Wen & Keyes, 2019; Semenoglou et al., 2023). There are also DA methods that take advantage of specific time series properties, e.g., window warping (Wen et al., 2021; Semenoglou et al., 2023), surrogate series (Keylock, 2006; Lee et al., 2019), and time-frequency feature augmentation (Keylock, 2006; Steven Eyobu & Han, 2018; Park et al., 2019; Gao et al., 2020). For the time series AD task, window cropping and window flipping are also often used. In addition, label expansion and amplitude/phase perturbations are introduced in (Gao et al., 2020).

With the above, one may consider directly applying these DA methods to the forecasting task to expand training data. We perform such experiments with several popular TSF models, and the results are shown in Table 1. As can be observed, these DA techniques tend to generate reduced forecasting accuracy. The reason is rather simple: most of these DA methods introduce perturbation in the time domain (e.g., noise or zero masking), and the augmented data-label pairs do not preserve the semantics consistency required by the TSF task.

There are also a few DA techniques for the forecasting task presented in the literature. (Hu et al., 2020) proposes *DATSING*, a transfer learning-based framework that leverages cross-domain time series latent representations to augment target domain forecasting. (Bandara et al., 2021) introduces two DA methods for forecasting: (i). Average selected with distance (ASD), which generates augmented time series using the weighted sum of multiple time series (Forestier et al., 2017), and the weights are determined by the dynamic time warping (DTW) distance; (ii). Moving block bootstrapping (MBB) generates augmented data by manipulating the residual part of the time series after STL decomposition (Cleveland et al., 1990; Semenoglou et al., 2023) and recombining it with the other series. It is worth noting that, in these works, data augmentation is not the focus of their proposed framework, and the design choices for their augmentation strategies are not thoroughly discussed. Further, a recent work (Semenoglou et al., 2023) proposed a linear upsampling-based method which is claimed to achieve performance boost on the commonly used forecasting benchmarks.

### 2.3 WHY FREQUENCY DOMAIN AUGMENTATION?

The forecastable behavior of time series data is usually driven by some periodical events. For example, the hourly sampled power consumption of a house is closely related to the periodical behavior of the householder. His/her daily routine (e.g., out for work during the day and activities at night) would introduce a daily periodicity, his/her routines between weekdays and weekends would introduce a weekly periodicity, while the yearly atmosphere temperature change would introduce an annual periodicity. Such periodical events can be easily decoupled in the frequency domain and manipulated independently.

Motivated by the above, we propose to perform frequency domain augmentation for the TSF task. It is worth noting that, while periodical events are decoupled in the frequency domain, augmentation methods still need careful design. As shown in Table1, simply introducing noise to frequency components could compromise the model's performance, as it would create a periodicity that does not exist in practice. In contrast, by identifying and manipulating events in the frequency domain for data points in both the look-back window and forecasting horizon, the resulting augmented data-label pair from *FrAug* could largely conform to the behavior of the underlying system.

## 3 METHODS

In this section, we detail the proposed frequency domain data augmentation methods for time series forecasting. We also present a detailed procedure of our methods in the Appendix.

### 3.1 THE PIPELINE OF FRAUG

To ensure the semantic consistency of augmented data-label pairs in forecasting, we add frequency domain perturbations on the concatenated time series of the look-back window and target horizon, with the help of the Fast Fourier transform (FFT) as introduced in Appendix. We only apply FrAug during the training stage and use original test samples for testing.

In the training stage, given a training sample (data points in the look-back window and the forecasting horizon), FrAug (i) concatenates the two parts, (ii) performs frequency domain augmentations, and (iii) splits the concatenated sequence back into lookback window and target horizon in the time domain. The augmentation result of an example time series training sample is shown in Figure 1.

### 3.2 FREQUENCY MASKING AND FREQUENCY MIXING

We propose two simple yet effective augmentation methods under FrAug framework, namely Frequency Masking and Frequency Mixing. Specifically, frequency masking randomly masks some frequency components, while frequency mixing exchanges some frequency components of two training samples in the dataset.

**Frequency masking:** The pipeline of frequency masking is shown in Figure 2(a). For a training sample comprised of data points in the look-back window $x_{t-b:t}$ and the forecasting horizon $x_{t+1:t+h}$, we first concatenate them in the time domain as $s = x_{t-b:t+h}$ and apply real FFT to calculate the frequency domain representation $S$, which is a tensor composed of the complex number. Next, we

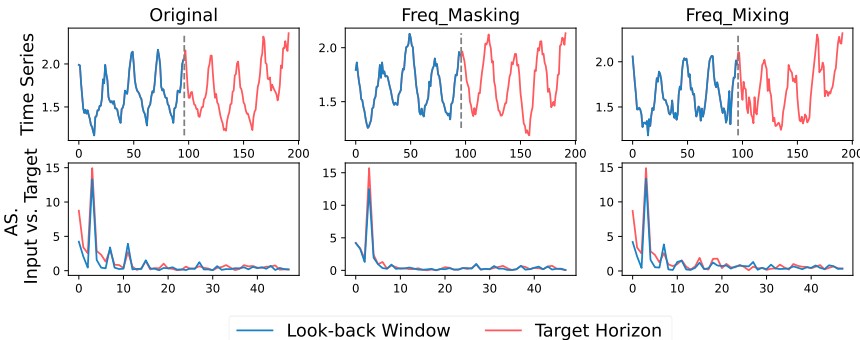

Figure 1: Visualization results of FrAug on 1000~1192 frames of 'OT' channel of ETTh1 dataset. The second row shows the original look-back window and target horizon have a similar distribution in amplitude spectrum (AS.). FrAug can preserve such consistency.

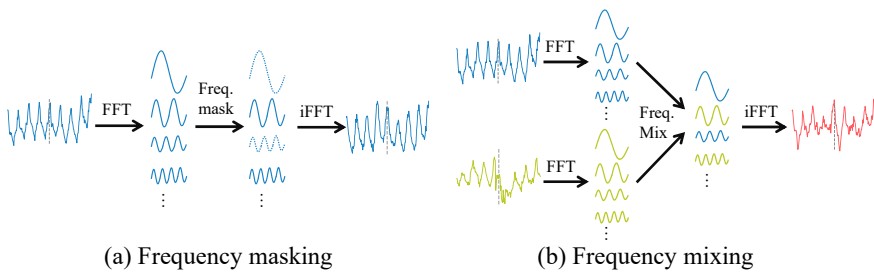

Figure 2: Illustration of the proposed augmentation methods.

randomly mask a portion of this complex tensor $S$ as zero and get $\tilde{S}$. Finally, we apply inverse real FFT to project the augmented frequency domain representation back to the time domain $\tilde{s} = \tilde{x}_{t-b:t+h}$.

Frequency Masking corresponds to removing some events in the underlying system. For example, considering the household power consumption time series, removing the weekly frequency components would create an augmented time series that belongs to a house owner that has similar activities on the weekdays and weekends.

**Frequency mixing:** The pipeline of frequency mixing is shown in Figure 2, wherein we randomly replace the frequency components in one training sample with the same frequency components of another training sample in the dataset.

Similarly, frequency mixing can be viewed as exchanging events between two samples. For the earlier example on household power consumption, the augmented time series could be one owner's weekly routine replaced by another's, and hence the augmented data-label pair largely preserves semantical consistency for forecasting.

Note that frequency masking and frequency mixing only utilize information from the original dataset, thereby avoiding the introduction of unexpected noises compared to those dataset expansion techniques based on synthetic generation (Esteban et al., 2017; Yoon et al., 2019). Moreover, as the number of combinations of training samples and their frequency components in a dataset is extremely large, FrAug can generate nearly infinite reasonable samples.

While frequency masking and frequency mixing are simple augmentation methods, we hereby emphasize the significance of the rationale behind them. These methods diverge significantly from traditional augmentation techniques as they would introduce substantial alterations to the time series data. Without the insights provided in this paper, one might find it counter-intuitive that such vastly-different augmented data can effectively contribute to model improvement.

Table 2: Comparison of different augmentation methods on ETT benchmarks under four forecasting lengths, including 96,192, 336, and 720. Performances are measured by MSE. The **best results** are highlighted in **bold**. **FrAug achieves the best performance in 80% of cases.**

| Model | Method | ETTh1 | | | | ETTh2 | | | | ETTm1 | | | | ETTm2 | | | |
|---|---|---|---|---|---|---|---|---|---|---|---|---|---|---|---|---|---|
| | | 96 | 192 | 336 | 720 | 96 | 192 | 336 | 720 | 96 | 192 | 336 | 720 | 96 | 192 | 336 | 720 |
| MICN | Original | 0.390 | 0.496 | 0.570 | 0.652 | 0.331 | 0.501 | 0.589 | 0.818 | 0.320 | 0.385 | 0.402 | 0.488 | 0.185 | 0.278 | 0.395 | 0.565 |
| | FreqMask | **0.388** | 0.487 | 0.521 | 0.604 | 0.310 | 0.478 | 0.610 | 0.832 | 0.313 | 0.357 | **0.389** | **0.469** | **0.178** | 0.266 | 0.413 | 0.540 |
| | FreqMix | **0.388** | 0.472 | **0.497** | **0.587** | **0.309** | 0.462 | **0.572** | 0.791 | **0.309** | **0.356** | 0.393 | 0.475 | 0.180 | **0.254** | **0.370** | **0.527** |
| | ASD | **0.388** | 0.475 | 0.565 | **0.587** | 0.337 | 0.574 | 0.764 | 0.831 | 0.317 | 0.367 | 0.402 | 0.477 | 0.194 | 0.334 | 0.434 | 0.574 |
| | MBB | 0.409 | **0.464** | 0.617 | 0.634 | 0.430 | 0.604 | 0.771 | 1.017 | 0.333 | 0.372 | 0.417 | 0.486 | 0.190 | 0.300 | 0.420 | 0.607 |
| | upsample | 0.432 | 0.515 | 0.554 | 0.640 | 0.334 | 0.481 | 0.660 | **0.767** | 0.348 | 0.388 | 0.406 | 0.491 | 0.180 | 0.275 | 0.399 | 0.552 |
| | Robusttad | 0.394 | 0.515 | 0.598 | 0.642 | 0.314 | 0.537 | 0.813 | 1.038 | 0.338 | 0.381 | 0.409 | 0.474 | 0.185 | 0.291 | 0.372 | 0.583 |
| Film | Original | 0.463 | 0.472 | 0.495 | 0.488 | 0.361 | 0.431 | 0.459 | 0.457 | 0.216 | 0.273 | 0.325 | 0.426 | 0.217 | 0.272 | 0.323 | 0.418 |
| | FreqMask | **0.451** | 0.465 | 0.492 | 0.484 | 0.349 | **0.420** | **0.448** | **0.443** | **0.212** | **0.270** | 0.317 | 0.414 | **0.210** | **0.268** | **0.316** | **0.413** |
| | FreqMix | 0.453 | **0.459** | **0.485** | 0.495 | 0.354 | 0.428 | 0.455 | 0.455 | **0.212** | **0.270** | 0.318 | **0.414** | 0.212 | 0.271 | 0.318 | 0.416 |
| | ASD | 0.456 | 0.514 | 0.498 | 0.521 | **0.340** | 0.451 | 0.505 | 0.452 | 0.447 | 0.466 | 0.546 | 0.590 | 0.212 | 0.276 | 0.343 | 0.427 |
| | MBB | 0.454 | 0.487 | 0.505 | 0.542 | 0.377 | 0.480 | 0.501 | 0.464 | 0.464 | 0.468 | 0.581 | 0.602 | 0.227 | 0.287 | 0.343 | 0.447 |
| | upsample | 0.454 | 0.508 | 0.519 | 0.515 | 0.369 | 0.426 | 0.450 | **0.443** | 0.511 | 0.487 | 0.560 | 0.606 | 0.215 | 0.288 | 0.345 | 0.430 |
| | Robusttad | 0.460 | 0.501 | 0.541 | **0.483** | 0.379 | 0.438 | 0.460 | 0.466 | 0.471 | 0.470 | 0.617 | 0.575 | 0.214 | 0.272 | 0.345 | 0.428 |
| DLinear | Original | 0.374 | **0.405** | 0.439 | 0.514 | 0.295 | 0.378 | 0.421 | 0.696 | 0.300 | 0.335 | **0.368** | **0.425** | 0.171 | 0.235 | 0.305 | 0.412 |
| | FreqMask | **0.372** | 0.407 | 0.453 | 0.473 | **0.282** | **0.344** | 0.443 | **0.592** | **0.297** | **0.332** | 0.368 | 0.428 | **0.166** | **0.228** | **0.281** | 0.399 |
| | FreqMix | **0.372** | 0.409 | 0.438 | 0.482 | 0.284 | 0.346 | 0.449 | 0.636 | **0.297** | **0.332** | 0.372 | 0.428 | 0.168 | 0.229 | 0.286 | **0.398** |
| | ASD | 0.387 | 0.554 | 0.445 | **0.467** | 0.302 | 0.363 | **0.411** | 0.677 | 0.311 | 0.343 | 0.377 | 0.430 | 0.188 | 0.237 | 0.297 | 0.400 |
| | MBB | 0.389 | 0.423 | 0.508 | 0.521 | 0.313 | 0.391 | 0.433 | 0.651 | 0.307 | 0.339 | 0.373 | 0.428 | 0.177 | 0.242 | 0.323 | 0.430 |
| | Upsample | 0.419 | 0.425 | **0.435** | 0.479 | 0.305 | 0.373 | 0.415 | 0.628 | 0.347 | 0.364 | 0.400 | 0.469 | 0.172 | 0.309 | 0.286 | 0.418 |
| | Robusttad | 0.373 | 0.408 | 0.461 | 0.473 | 0.291 | 0.393 | 0.424 | 0.760 | 0.300 | 0.336 | 0.373 | 0.427 | 0.169 | 0.231 | 0.300 | 0.427 |
| Fedformer | Original | 0.374 | 0.425 | 0.456 | 0.485 | 0.339 | 0.430 | 0.519 | 0.474 | 0.364 | 0.406 | **0.446** | 0.533 | 0.189 | 0.253 | 0.327 | 0.438 |
| | FreqMask | 0.374 | 0.421 | 0.457 | **0.474** | **0.323** | **0.414** | 0.484 | 0.446 | **0.360** | **0.404** | 0.453 | 0.527 | **0.184** | **0.249** | **0.322** | 0.430 |
| | FreqMix | **0.371** | **0.418** | 0.466 | 0.497 | 0.327 | 0.421 | 0.507 | 0.466 | 0.361 | **0.404** | 0.447 | **0.515** | **0.184** | 0.252 | 0.326 | 0.432 |
| | ASD | 0.429 | 0.455 | 0.561 | 0.582 | 0.339 | 0.429 | 0.501 | 0.454 | 0.390 | 0.430 | 0.514 | 0.585 | 0.200 | 0.264 | 0.345 | 0.460 |
| | MBB | 0.412 | 0.460 | 0.501 | 0.514 | 0.356 | 0.455 | 0.526 | 0.484 | 0.385 | 0.427 | 0.477 | 0.548 | 0.211 | 0.270 | 0.340 | 0.439 |
| | Upsample | 0.400 | 0.438 | 0.460 | 0.463 | 0.327 | 0.416 | **0.473** | **0.442** | 0.441 | 0.470 | 0.544 | 0.545 | 0.194 | 0.265 | 0.330 | **0.422** |
| | Robusttad | 0.377 | 0.424 | **0.453** | 0.479 | 0.348 | 0.435 | 0.501 | 0.474 | 0.375 | 0.413 | 0.455 | **0.515** | 0.190 | 0.261 | 0.324 | 0.431 |
| Autoformer | Original | 0.449 | 0.463 | 0.495 | 0.535 | 0.432 | 0.430 | 0.482 | 0.471 | 0.552 | 0.559 | 0.605 | 0.755 | 0.288 | 0.274 | 0.335 | 0.437 |
| | FreqMask | 0.434 | **0.426** | 0.495 | 0.597 | **0.347** | 0.425 | 0.474 | **0.465** | 0.419 | 0.513 | **0.481** | **0.595** | 0.232 | **0.264** | **0.325** | **0.421** |
| | FreqMix | **0.401** | 0.484 | **0.471** | 0.517 | 0.351 | **0.423** | 0.494 | 0.538 | **0.410** | 0.542 | 0.497 | 0.769 | **0.212** | 0.265 | **0.325** | 0.433 |
| | ASD | 0.486 | 0.497 | 0.530 | **0.499** | 0.362 | 0.442 | 0.477 | 0.523 | 0.561 | 0.532 | 0.518 | 0.616 | 0.233 | 0.276 | 0.331 | 0.444 |
| | MBB | 0.479 | 0.526 | 0.592 | 0.602 | 0.363 | 0.431 | 0.472 | 0.547 | 0.535 | 0.652 | 0.704 | 0.822 | 0.239 | 0.283 | 0.334 | 0.454 |
| | Upsample | 0.416 | 0.578 | 0.484 | 0.506 | 0.370 | 0.454 | **0.459** | 0.493 | 0.558 | 0.651 | 0.585 | 0.654 | 0.245 | 0.275 | 0.333 | 0.461 |
| | Robusttad | 0.467 | 0.456 | 0.492 | **0.503** | 0.420 | 0.503 | 1.023 | 0.536 | 0.495 | **0.505** | 0.546 | 0.669 | 0.295 | 0.272 | 0.333 | 0.434 |
| Informer | Original | 0.931 | 1.010 | 1.036 | 1.159 | 2.843 | 6.236 | 5.418 | 3.962 | 0.626 | 0.730 | 1.037 | 0.972 | 0.389 | 0.813 | 1.429 | 3.863 |
| | FreqMask | **0.637** | **0.788** | **0.873** | **1.042** | 2.555 | 3.983 | **3.752** | **2.561** | **0.425** | **0.538** | **0.783** | **0.846** | 0.368 | **0.463** | **0.984** | 3.932 |
| | FreqMix | 0.675 | 1.021 | 1.044 | 1.100 | 2.774 | 5.940 | 4.718 | 4.088 | 0.593 | 0.702 | 0.867 | 0.930 | **0.341** | 0.557 | 1.322 | 2.939 |
| | ASD | 0.853 | 1.020 | 1.124 | 1.226 | 2.280 | 5.830 | 4.345 | 3.886 | 0.726 | 0.775 | 0.921 | 0.968 | 0.405 | 0.874 | 1.317 | 2.585 |
| | MBB | 0.958 | 1.031 | 1.049 | 1.227 | 3.112 | 6.398 | 5.668 | 4.007 | 0.630 | 0.731 | 0.988 | 0.961 | 0.399 | 0.783 | 1.476 | 4.012 |
| | Upsample | 1.362 | 1.040 | 1.056 | 1.161 | **1.375** | **1.852** | 4.004 | 3.073 | 0.769 | 0.791 | 1.035 | 1.016 | 0.383 | **0.463** | 1.071 | **1.874** |
| | Robusttad | 1.044 | 1.018 | 1.258 | 1.169 | 3.120 | 5.387 | 5.518 | 3.698 | 0.652 | 0.751 | 1.206 | 1.099 | 0.346 | 0.869 | 1.338 | 3.612 |

## 4 EXPERIMENTS

In this section, we apply FrAug to three TSF application scenarios: long-term forecasting, cold-start forecasting, and test-time training. Due to space limitations, the experimental results for other applications (e.g., short-term forecasting) and ablation studies are presented in the Appendix.

### 4.1 EXPERIMENTAL SETUP

**Dataset**. All datasets used in our experiments are widely-used and publicly available real-world datasets, including Exchange-Rate (Lai et al., 2017), Traffic, Electricity, Weather, ILI, ETT (Zhou et al., 2021). We summarize the characteristics of these datasets in the appendix.

**Baselines**. We compare FrAug, including Frequency Masking (FreqMask) and Frequency Mixing (FreqMix), with existing time-series augmentation techniques, including ASD (Forestier et al., 2017; Bandara et al., 2021), MBB (Bandara et al., 2021; Bergmeir et al., 2016; Semenoglou et al., 2023) and Upsampling (Semenoglou et al., 2023), and Robusttad (Gao et al., 2020).

**Deep Models**. We include six state-of-the-art deep learning models for long-term forecasting, including Informer (Zhou et al., 2021), Autoformer (Wu et al., 2021), FEDformer (Zhou et al., 2022a), DLinear (Zeng et al., 2022), Film (Zhou et al., 2022b) and MICN (Wang et al., 2022). The effectiveness of augmentation methods is evaluated by comparing the performance of the same model trained with different augmentations.

Table 3: Comparison on four datasets. Performances are measured by MSE. The **best results** are highlighted in **bold**. **FrAug achieves the best performance in 74% of cases.**

| Model | Method | Exchange Rate | | | | Electricity | | | | Traffic | | | | Weather | | | |
|---|---|---|---|---|---|---|---|---|---|---|---|---|---|---|---|---|---|
| | | 96 | 192 | 336 | 720 | 96 | 192 | 336 | 720 | 96 | 192 | 336 | 720 | 96 | 192 | 336 | 720 |
| MICN | Original | 0.107 | 0.198 | 0.355 | 0.814 | 0.175 | 0.188 | 0.199 | 0.248 | 0.541 | 0.564 | 0.576 | 0.667 | 0.165 | 0.228 | 0.265 | 0.340 |
| | FreqMask | 0.106 | 0.185 | 0.309 | 0.739 | 0.147 | 0.161 | 0.173 | 0.224 | **0.496** | 0.518 | 0.534 | **0.615** | **0.164** | 0.213 | 0.271 | 0.334 |
| | FreqMix | 0.106 | 0.186 | 0.302 | 0.707 | **0.146** | **0.158** | **0.169** | **0.221** | 0.498 | **0.513** | **0.530** | 0.616 | 0.168 | **0.211** | 0.261 | 0.336 |
| | ASD | 0.100 | 0.188 | 0.333 | 0.925 | 0.165 | 0.170 | 0.191 | 0.234 | 0.535 | 0.551 | 0.568 | 0.628 | 0.165 | 0.228 | **0.258** | 0.329 |
| | MBB | 0.102 | 0.193 | 0.357 | 1.006 | 0.180 | 0.186 | 0.200 | 0.249 | 0.531 | 0.554 | 0.565 | 0.647 | 0.170 | 0.227 | 0.266 | 0.342 |
| | upsample | 0.103 | **0.184** | **0.280** | **0.554** | 0.168 | 0.185 | 0.198 | 0.248 | 0.540 | 0.575 | 0.596 | 0.682 | 0.196 | 0.241 | 0.307 | 0.362 |
| | Robusttad | **0.097** | 0.185 | 0.320 | 0.814 | 0.189 | 0.192 | 0.202 | 0.241 | 0.535 | 0.552 | 0.568 | 0.650 | 0.183 | 0.225 | 0.282 | **0.333** |
| Film | Original | 0.140 | 0.294 | 0.440 | 1.103 | 0.185 | **0.204** | 0.225 | 0.324 | 0.630 | 0.648 | **0.652** | **0.663** | 0.250 | 0.297 | 0.345 | 0.418 |
| | FreqMask | **0.139** | 0.288 | 0.436 | 1.091 | 0.185 | 0.208 | **0.223** | 0.304 | 0.627 | 0.647 | 0.655 | 0.666 | **0.218** | **0.277** | **0.323** | 0.387 |
| | FreqMix | 0.140 | 0.293 | 0.439 | 1.108 | **0.184** | **0.204** | 0.225 | 0.295 | **0.621** | **0.639** | 0.655 | 0.667 | 0.245 | 0.296 | 0.350 | 0.390 |
| | ASD | 0.151 | 0.267 | 0.457 | 1.153 | 0.207 | 0.210 | 0.226 | **0.253** | 0.782 | 0.658 | 0.680 | 0.695 | 0.216 | 0.283 | 0.331 | 0.388 |
| | MBB | 0.152 | 0.267 | 0.470 | 1.107 | 0.198 | 0.217 | 0.225 | 0.278 | 0.634 | 0.662 | 0.637 | 0.679 | 0.257 | 0.323 | 0.360 | 0.436 |
| | upsample | 0.149 | **0.255** | **0.388** | **1.055** | 0.209 | 0.230 | 0.329 | 0.356 | 0.778 | 0.801 | 0.777 | 0.777 | 0.258 | 0.332 | 0.334 | **0.384** |
| | Robusttad | 0.170 | 0.268 | 0.413 | 1.075 | 0.203 | 0.211 | 0.227 | 0.297 | 0.632 | 0.644 | 0.659 | 0.697 | 0.239 | 0.302 | 0.356 | 0.409 |
| DLinear | Original | **0.079** | 0.205 | 0.309 | 1.029 | **0.140** | **0.154** | **0.169** | **0.204** | **0.410** | **0.423** | 0.436 | **0.466** | 0.175 | 0.217 | 0.265 | **0.324** |
| | FreqMask | 0.101 | **0.182** | 0.263 | 0.842 | **0.140** | **0.154** | **0.169** | **0.204** | 0.411 | **0.423** | **0.435** | **0.466** | **0.174** | 0.217 | 0.265 | **0.324** |
| | FreqMix | 0.099 | 0.187 | 0.274 | 0.883 | **0.140** | **0.154** | **0.169** | **0.204** | 0.412 | **0.423** | **0.435** | 0.467 | **0.174** | **0.216** | 0.264 | **0.324** |
| | ASD | 0.102 | 0.273 | 0.294 | 0.787 | 0.163 | 0.175 | 0.189 | 0.222 | 0.437 | 0.450 | 0.463 | 0.493 | 0.195 | 0.230 | 0.275 | 0.329 |
| | MBB | 0.080 | 0.204 | 0.308 | 1.021 | 0.145 | 0.157 | 0.172 | 0.206 | 0.420 | 0.430 | 0.441 | 0.467 | 0.176 | 0.217 | **0.262** | **0.324** |
| | Upsample | 0.097 | 0.224 | **0.254** | **0.571** | 0.149 | 0.162 | 0.176 | 0.209 | 0.445 | 0.454 | 0.466 | 0.492 | 0.185 | 0.234 | 0.273 | 0.337 |
| | Robusttad | 0.080 | 0.185 | 0.330 | 1.011 | **0.140** | 0.155 | **0.169** | **0.204** | 0.412 | 0.426 | 0.440 | **0.466** | **0.174** | **0.216** | 0.263 | 0.332 |
| Fedformer | Original | 0.135 | 0.271 | 0.454 | 1.140 | 0.188 | 0.196 | 0.212 | 0.250 | 0.574 | 0.611 | 0.623 | 0.631 | 0.250 | 0.266 | 0.368 | 0.397 |
| | FreqMask | **0.129** | **0.238** | 0.469 | 1.149 | **0.176** | 0.196 | **0.204** | **0.220** | 0.572 | **0.586** | 0.612 | 0.631 | 0.231 | **0.240** | **0.308** | **0.373** |
| | FreqMix | 0.133 | 0.241 | 0.470 | 1.147 | **0.176** | **0.187** | **0.204** | 0.226 | **0.565** | 0.591 | **0.604** | 0.629 | **0.200** | 0.245 | 0.317 | 0.377 |
| | ASD | 0.149 | 0.265 | 0.441 | 1.128 | 0.192 | 0.205 | 0.214 | 0.243 | 0.573 | 0.601 | 0.608 | **0.613** | 0.700 | 0.513 | 0.623 | 0.649 |
| | MBB | 0.145 | 0.257 | 0.456 | 1.142 | 0.202 | 0.223 | 0.240 | 0.297 | 0.601 | 0.613 | 0.630 | 0.647 | 0.309 | 0.280 | 0.352 | 0.392 |
| | Upsample | 0.155 | 0.259 | **0.426** | **1.009** | 0.199 | 0.213 | 0.224 | 0.247 | 0.644 | 0.656 | 0.657 | 0.675 | 0.225 | 0.276 | 0.326 | 0.401 |
| | Robusttad | 0.159 | 0.269 | 0.453 | 1.176 | 0.188 | 0.194 | 0.211 | 0.228 | 0.569 | 0.620 | 0.620 | 0.631 | 0.227 | 0.492 | 0.610 | 0.400 |
| Autoformer | Original | 0.145 | 0.385 | **0.453** | 1.087 | 0.203 | 0.231 | 0.247 | 0.276 | 0.624 | 0.619 | 0.604 | 0.703 | 0.271 | 0.315 | 0.345 | 0.452 |
| | FreqMask | **0.139** | 0.414 | 0.838 | **0.806** | 0.170 | 0.209 | 0.212 | **0.237** | 0.594 | **0.588** | 0.608 | 0.654 | **0.217** | **0.280** | 0.323 | 0.388 |
| | FreqMix | 0.155 | 0.668 | 0.615 | 2.093 | **0.163** | **0.189** | **0.204** | 0.243 | **0.559** | 0.618 | **0.583** | 0.649 | 0.252 | 0.302 | 0.334 | 0.388 |
| | ASD | 0.147 | 0.312 | 1.344 | 1.152 | 0.248 | 0.223 | 0.268 | 0.254 | 0.608 | 0.616 | 0.603 | 0.694 | 1.015 | 0.574 | 0.584 | 0.874 |
| | MBB | 0.152 | **0.273** | 0.472 | 1.641 | 0.231 | 0.317 | 0.269 | 0.272 | 0.628 | 0.650 | 0.658 | 0.664 | 0.237 | 0.349 | 0.376 | 0.451 |
| | Upsample | 0.142 | 0.331 | 0.532 | 1.033 | 0.199 | 0.232 | 0.247 | 0.301 | 0.661 | 0.835 | 0.668 | 1.012 | 0.334 | 0.295 | **0.314** | **0.377** |
| | Robusttad | 0.144 | 0.326 | 0.969 | 1.178 | 0.321 | 0.424 | 0.446 | 0.264 | 0.621 | 0.602 | 0.594 | 0.643 | 0.322 | 0.342 | 0.343 | 0.421 |
| Informer | Original | 0.879 | 1.147 | 1.562 | 2.919 | 0.305 | 0.349 | 0.349 | 0.391 | 0.736 | 0.770 | 0.861 | 0.995 | 0.452 | 0.466 | 0.499 | 1.260 |
| | FreqMask | **0.534** | 1.023 | **1.074** | **1.102** | **0.262** | 0.282 | **0.287** | **0.304** | **0.674** | 0.683 | **0.715** | 0.799 | 0.199 | 0.298 | **0.356** | **0.529** |
| | FreqMix | 0.962 | 1.156 | 1.514 | 2.689 | 0.266 | **0.276** | **0.287** | 0.306 | **0.674** | **0.677** | 0.726 | 0.807 | 0.216 | 0.402 | 0.459 | 0.666 |
| | ASD | 0.994 | 1.132 | 1.669 | 1.924 | 0.317 | 0.331 | 0.334 | 0.348 | 0.812 | 0.747 | 0.805 | 0.900 | 0.342 | 0.452 | 0.529 | 0.644 |
| | MBB | 0.859 | 1.136 | 1.549 | 2.874 | 0.354 | 0.389 | 0.397 | 0.451 | 0.752 | 0.768 | 0.892 | 1.057 | 0.544 | 0.425 | 0.601 | 1.221 |
| | Upsample | 0.847 | **0.917** | 1.442 | 2.007 | 0.375 | 0.361 | 0.391 | 0.366 | 0.738 | 0.817 | 0.935 | 0.972 | 0.531 | 0.536 | 0.634 | 0.902 |
| | Robusttad | 0.998 | 1.216 | 1.732 | 2.971 | 0.308 | 0.324 | 0.331 | 0.342 | 0.710 | 0.721 | 0.791 | 0.872 | 0.349 | 0.622 | 0.654 | 1.099 |

**Evaluation metrics**. Following previous works (Zhou et al., 2021; Wu et al., 2021; Zhou et al., 2022a; Zeng et al., 2022; Zhang et al., 2022; Zhou et al., 2022b; Liu et al., 2022), we use Mean Squared Error (MSE) as the core metrics to compare forecasting performance.

**Implementation**. FreqMask and FreqMix only have one hyper-parameter, which is the mask rate/mix rate, respectively. In our experiments, we only consider 0.1, 0.2,0.3,0.4,0.5. We use cross-validation to select this hyper-parameter. More details are presented in the Appendix.

## 4.2 LONG-TERM FORECASTING

Table 2 and Table 3 show the results of long-term forecasting. We use different augmentation methods to double the size of the training dataset.

As can be observed, FreqMask and FreqMix improve the performance of the original model and outperform other methods in 77% of cases, while applying data augmentation with ASD, MBB and Robusttad are often inferior to the original model. Upsample sometimes achieves comparable results with FrAug, especially for the exchange rate dataset. This particular dataset does not have apparent periodicities. Therefore, its main forecasting behavior can be viewed as identifying the trend. From this perspective, the upsample method that linearly interpolates time series can create more reasonable augmentations compared to FrAug under such circumstances. However, since upsample would change the periodicity of data, it lags behind FrAug consistently for those datasets with steady periodicity, such as electricity, traffic, and weather.

Notably, FreqMask improves DLinear's performance by 16% in ETTh2 when the predicted length is 192, and it improves FEDformer's performance by 28% and Informer's performance by 56% for the Weather dataset when the predicted length is 96. Similarly, FreqMix improves the performance of Autoformer by 27% for ETTm2 with a predicted length of 96 and the performance of Informer

Table 4: Performance of models trained with the last 1% training samples compared with that trained with the full training set. Performances are measured by MSE. The **best results** are highlighted in **bold** (row full data is not included). **FrAug achieves the best performance in 97% of cases.**

| Model | Method | ETTh2 | | | | Exchange Rate | | | | Electricity | | | | Traffic | | | |
|---|---|---|---|---|---|---|---|---|---|---|---|---|---|---|---|---|---|
| | | 96 | 192 | 336 | 720 | 96 | 192 | 336 | 720 | 96 | 192 | 336 | 720 | 96 | 192 | 336 | 720 |
| DLinear | Full Data | 0.295 | 0.378 | 0.421 | 0.696 | 0.079 | 0.205 | 0.309 | 1.029 | 0.140 | 0.154 | 0.169 | 0.204 | 0.410 | 0.423 | 0.436 | 0.466 |
| | 1% Data | 0.572 | 0.704 | 0.628 | 0.662 | 0.280 | 0.549 | 1.711 | 0.897 | 0.196 | 0.205 | 0.218 | 0.280 | 0.764 | 0.658 | 0.825 | 0.908 |
| | FreqMask | **0.351** | **0.467** | **0.541** | **0.640** | 0.174 | 0.258 | **0.777** | **0.820** | **0.172** | **0.183** | **0.197** | **0.257** | **0.466** | **0.484** | **0.509** | **0.539** |
| | FreqMix | 0.464 | 0.531 | 0.590 | 1.005 | **0.139** | **0.252** | 1.205 | 0.897 | 0.176 | 0.185 | 0.200 | **0.257** | 0.486 | 0.510 | 0.515 | 0.578 |
| | ASD | 1.607 | 1.626 | 2.071 | 6.734 | 0.396 | 0.793 | 1.254 | 2.380 | 0.204 | 0.211 | 0.225 | 0.281 | 0.516 | 0.541 | 0.582 | 0.575 |
| | MBB | 1.834 | 1.702 | 1.789 | 7.008 | 0.425 | 0.869 | 1.290 | 2.411 | 0.211 | 0.218 | 0.230 | 0.285 | 0.583 | 0.620 | 0.630 | 0.639 |
| | Upsample | 1.696 | 2.338 | 1.974 | 0.670 | 0.325 | 0.513 | 1.029 | 1.264 | 0.218 | 0.223 | 0.234 | 0.288 | 0.549 | 0.590 | 0.610 | 0.600 |
| | Robusttad | 1.519 | 1.808 | 1.852 | 6.740 | 0.393 | 0.789 | 1.253 | 2.379 | 0.203 | 0.210 | 0.223 | 0.282 | 0.535 | 0.559 | 0.578 | 0.575 |
| Autoformer | Full Data | 0.432 | 0.430 | 0.482 | 0.471 | 0.145 | 0.385 | 0.453 | 1.087 | 0.203 | 0.231 | 0.247 | 0.276 | 0.624 | 0.619 | 0.604 | 0.703 |
| | 1% Data | 0.490 | 0.546 | 0.498 | 0.479 | 0.247 | 0.487 | **0.579** | 1.114 | 0.462 | 0.488 | 0.518 | 0.541 | 1.238 | 1.366 | 1.299 | 1.325 |
| | FreqMask | **0.409** | **0.496** | **0.476** | **0.471** | **0.166** | **0.304** | 0.791 | **1.085** | **0.326** | 0.331 | 0.410 | 0.483 | **0.761** | **0.832** | **0.706** | **0.786** |
| | FreqMix | 0.414 | 0.502 | 0.484 | **0.471** | 0.208 | 0.369 | 0.832 | 1.091 | 0.347 | **0.326** | **0.381** | **0.479** | **0.746** | 0.844 | 0.715 | 0.795 |
| | MBB | 0.443 | 0.460 | 0.481 | 0.487 | 0.304 | 0.400 | 0.600 | 1.259 | 0.398 | 0.428 | 0.453 | 0.506 | 0.761 | 1.048 | 1.044 | 1.048 |
| | ASD | 0.468 | 0.512 | 0.480 | 0.481 | 0.226 | 0.373 | 0.588 | 1.267 | 0.349 | 0.424 | 0.420 | 0.513 | 0.907 | 1.034 | 0.784 | 0.836 |
| | Upsample | 0.425 | 0.492 | 0.473 | 0.468 | 0.173 | 0.269 | 0.607 | 1.263 | 0.422 | 0.477 | 0.516 | 0.544 | 1.079 | 1.099 | 1.110 | 1.130 |
| | Robusttad | 0.455 | 0.526 | 0.492 | 0.480 | 0.298 | 0.476 | 0.813 | 1.21 | 0.383 | 0.441 | 0.510 | 0.520 | 0.808 | 0.920 | 1.046 | 1.053 |

by 35% for ETT2 with a predicted length of 720. These results indicate that FrAug is an effective DA solution for long-term forecasting, significantly boosting SOTA performance in many cases. We attribute the performance improvements brought by FrAug to its ability to alleviate the over-fitting issues. Limited by space, we provide further analysis in the Appendix.

## 4.3 COLD-START FORECASTING

Another important application of FrAug is cold-start forecasting, where only very few training samples are available.

To simulate this scenario, we reduce the number of training samples of each dataset to the last 1% of the original size. For example, we only use the 8366th-8449th training samples (83 in total) of the ETTh2 dataset for training. Then, we use FrAug to generate augmented data to enrich the dataset. In this experiment, we only consider two extreme cases: enlarging the dataset size to 2x or 50x, and the presented result is selected by cross-validation. Models are evaluated on the same test dataset as the original long-term forecasting tasks.

Table 4 shows the results of two forecasting models: DLinear and Autoformer on 4 datasets. The other results are shown in Appendix. We also include the results of models trained with the full dataset for comparison. As can be observed, FrAug outperforms all baseline methods consistently and maintains the overall performances of the models to a large extent. For the Traffic dataset, compared with the one trained on full training data, the overall performance drop of DLinear is 13% with FrAug, while the drop is 45% without FrAug. Surprisingly, sometimes the model performance is even better than those trained with the full dataset. For example, the performance of Autoformer is 5.4% better than the one trained with the full dataset for ETTh2 when the predicted length is 96. The performance of DLinear is 20% better than the one trained with the full dataset for Exchange Rate when the predicted length is 720. We attribute it to the distribution shift in the full training dataset, which could deteriorate the model performance.

## 4.4 TEST-TIME TRAINING

Distribution shifts often exist between the training data and the test data. For instance, as shown in Figure 4, the distribution of data (trend and mean value) changes dramatically at the middle and the end of the ILI dataset.

As time-series data is continuously generated in real applications, a natural solution to tackle this problem is to update the model when the new data is available for training (Pham et al., 2022). To simulate the real re-training scenario, we design a simple test-time training policy, where we divide the dataset into several parts according to their arrival times. The model is retrained after all the data points in each part are revealed. Specifically, the model is trained with the data in the first part and tested on data in the second part. Then, the model will be re-trained with data in both the first and second parts and tested on data in the third part.

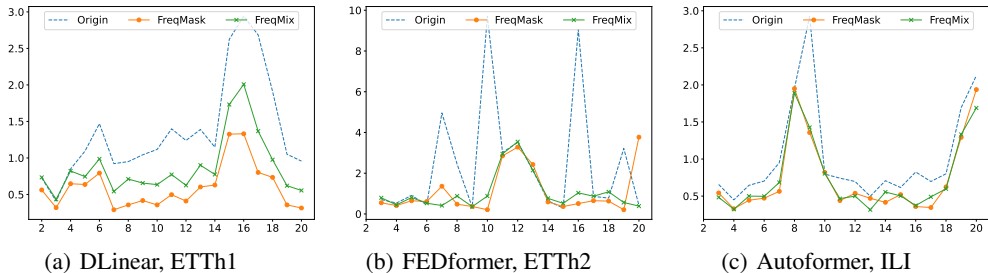

(a) DLinear, ETTh1      (b) FEDformer, ETTh2      (c) Autoformer, ILI

Figure 3: Visualization of the forecasting curves of different methods in every part of the dataset under test-time training policy. We divide a dataset into 20 parts. The X axis is the index of each part, and the Y axis shows the corresponding test error. When a distribution shift happens, the forecasting error increases significantly. FrAug can mitigate such performance degradation.

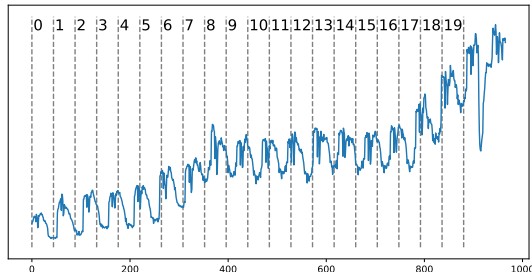

Figure 4: Visualization of the last channel of the ILI dataset. Distribution shifts happen at the middle and the end of the dataset.

In our experiments, we divide the dataset into 20 parts and use the above setting as the baseline. Then, we validate the effectiveness of FrAug under the same setting. FrAug can help the model fit into a new distribution by creating augmented samples with the new distribution. Specifically, for data that are just available for training, which is more likely to have the same distribution as future data, we create 5 augmented samples. The number of augmented samples gradually decreases to 1 for old historical data. We conduct experiments on ETTh1, ETTh2, and ILI datasets, which have relatively higher distribution shifts compared to other datasets. Figure 3 shows the test loss of the models in every part of the dataset. As expected, when a distribution shift happens, the test loss increases. However, FrAug can mitigate such performance degradation of the model under distribution shift. Notably, in Figure 3(c), when the distribution shift happens in the 8th part (refer to Figure 4 for visualization). With FrAug, the test loss decreases in the 9th part, while the test loss of the baseline solution keeps increasing without FrAug. This indicates that FrAug helps the model fit into the new distribution quickly. We present more quantitative results in the Appendix.

## 5 CONCLUSION

This work explores effective data augmentation techniques for the time series forecasting task. By systematically analyzing existing augmentation methods for time series, we first show that they are not applicable for TSF, as the augmented data-label pairs cannot meet the semantical consistency requirements in forecasting. Then, we propose FrAug, an easy-to-implement frequency domain augmentation solution, including frequency masking and frequency mixing strategies that effectively expand training data size.

Comprehensive experiments on widely used datasets validate that FrAug alleviates the overfitting problems of state-of-the-art TSF models, thereby improving their forecasting performance. In particular, we show that FrAug enables models trained with 1% of the original training data to achieve similar performance to the ones trained on full training data, making it an attractive solution for cold-start forecasting. Moreover, we show that applying test-time training with FrAug greatly improves forecasting accuracy for time series with significant distribution shifts, which often occurs in real-life TSF applications.

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

## A  APPENDIX

In this appendix, we provide 1) a brief introduction to the FFT in Sec. A.1, 2) detail algorithms of FrAug in Sec. A.2, 3) more implementation details in Sec. A.3, 4) an extra study of FrAug in short-term forecasting tasks in Sec. A.4, 5) more experiment results of FrAug in cold-start forecasting problem in Sec. A.6, 6) more experiment results of FrAug in time-time training problem in Sec. A.7. 6) ablation study of FrAug in Sec. A.8.

### A.1  DISCRETE FOURIER TRANSFORM AND FAST FOURIER TRANSFORM (FFT)

Specifically, DFT converts a finite sequence of equally-spaced samples of a function into a same-length sequence of equally-spaced samples of a complex-valued function of frequency. Given a sequence $x = \{x_n\}$ with $n \in [0, N-1]$, the DFT is defined by:

$$X_k = \sum_{n=0}^{N-1} x_n e^{-\frac{2\pi i}{N} nk}, 0 \le k \le N-1$$

The most commonly-used DFT calculating method is Fast Fourier Transform (FFT). However, when dealing with real number input, the positive and negative frequency parts are conjugate with each other. Thus, we can get a more compact one-sided representation where only the positive frequencies are preserved, which have length of $(N+1)//2$. We use *pyTorch* function *torch.fft.rfft* and *torch.fft.irfft* to perform real FFT and inverse real FFT. In the following subsections, we refer spectrum as the positive frequency spectrum calculated by real FFT.

### A.2  DETAIL ALGORITHM FOR FREQMASK AND FREQMIX

---
**Algorithm 1** Frequency Masking
---
**Input:** Look-back window $x$, target horizon $y$, mask rate $\mu$
**Output:** Augmented Look-back window $\tilde{x}$, augmented target horizon $\tilde{y}$
 1:  $s = x||y$; {Concatenate x and y}
 2:  $S = rFFT(s)$; {Calculate the frequency representation $S$. $S$ is composed of complex numbers and have length of $(b+h)//2 + 1$}
 3:  $m = CreateRandomMask(len(S), \mu)$ {Create random mask for frequency representation with mask rate $\mu$}
 4:  $\tilde{S} = Masking(S, m)$;
 5:  $\tilde{s} = irFFT(\tilde{S})$;
 6:  $\tilde{x}, \tilde{y} = s[0:b], s[b:b+t]$; {Split the augmented training sample}
---

---
**Algorithm 2** Frequency Mixing
---
**Input:** Look-back window $x1$, target horizon $y1$, another training sample pair $x2$, $y2$, mix rate $\mu$
**Output:** Augmented look-back window $\tilde{x}$, augmented target horizon $\tilde{y}$
 1:  $s1 = x1||y1, s2 = x2||y2$; {Concatenate x and y}
 2:  $S1 = rFFT(s1), S2 = rFFT(s2)$; {Calculate the frequency representation $S$. $S$ is composed of complex numbers and have length of $(b+h)//2 + 1$}
 3:  $m1 = CreateRandomMask(len(S), \mu)$ {Create random mask for frequency representation with mix rate $\mu$ no more than 0.5}
 4:  $m2 = BitwiseNOT(m1)$ {Create inverted mask for training sample 2}
 5:  $\tilde{S} = Masking(S1, m1) + Masking(S2, m2)$;
 6:  $\tilde{s} = irFFT(\tilde{S})$;
 7:  $\tilde{x}, \tilde{y} = s[0:b], s[b:b+t]$; {Split the augmented training sample}
---

We propose two simple yet effective augmentation methods under FrAug framework, namely Frequency Masking and Frequency Mixing. Specifically, frequency masking randomly masks some frequency components, while frequency mixing exchanges some frequency components of two

training samples in the dataset. Algorithm 1 and Algorithm 2 show the details process of FreqMask and FreqMix respectfully.

## A.3 IMPLEMENTATION DETAILS

Table 5: The statistics of the nine used datasets.

| Dataset | Exchange-Rate | Traffic | Electricity | Weather | ETTh1&2 | ETTm1 &2 | ILI |
|---------|---------------|---------|-------------|---------|---------|----------|-----|
| Variates | 8 | 862 | 321 | 21 | 7 | 7 | 7 |
| Frequency | 1day | 1hour | 1hour | 10min | 1hour | 5min | 1week |
| Timesteps | 7,588 | 17,544 | 26,304 | 52,696 | 17,420 | 69,680 | 966 |

**Dataset**. In Table 5, we provide basic information on the dataset used in this paper.

**FrAug**. In application, FrAug can be implemented by a few lines of code. In long-term forecasting and cold-start forecasting, when we double the size of the dataset, we apply FrAug in a batch-wise style. For example, when the experiments' default batch size is 32, we reduce it to 16. In the training process, we use FrAug to create an augmented version for each sample in the batch, so that the batch size is back to 32 again. Such a procedure can reduce memory costs since we don't need to generate and store all augmented samples at once. Also, such a design enhances the diversity of samples, since the mask/mix rate in FrAug introduces randomness to the augmentation. In test-time training and 50x augmentation in cold-start forecasting, we simply enlarge the size of the dataset before training.

**Implementation of baselines**. ASD (Forestier et al., 2017; Bandara et al., 2021) and MBB (Bandara et al., 2021; Bergmeir et al., 2016; Semenoglou et al., 2023) are reproduced by us based on the descriptions in their original paper. Specifically, for ASD, we first calculate the pair-wise DTW distances of all training samples. Then, to generate a new sample, we applied an exponentially weighted sum to each sample's top5 closest neighbors. This weighted sum is on both look-back window and horizon. Finally, we combine all new samples with the original dataset and train the model with them. For MBB, we apply a similar batch-wise augmentation procedure as FrAug. For each sample, we use the STL decompose from package *statsmodels* to extract the residual component of each training sample. Then we use the MovingBlockBootstrap from package *arch* to add perturbations on the residual component. Finally, we recombine the residual part with the other components. Robusttad Gao et al. (2020) adds Gaussian noise to the frequency components of a time series. This method is designed for anomaly detection tasks. Since it also augments time series in the frequency domain, we think that a comparison with this method is necessary.

**Implementation of Table 1**. In Table 1, we adapt several augmentation methods to the TSF task. The overall work flow of every method is the same as we mentioned in Sec 3.1. The implementation of Freq-Noise is similar to Robusttad Gao et al. (2020).

**Models**. For every experiment, we follow the hyper-parameters provided by the original repository of the models.

## A.4 SHORT-TERM FORECASTING TASKS

We present the results of FrAug on the short-term forecasting tasks in Table 6. For FEDformer, Autoformer and Informer, FrAug consistently improves their performances. Notably, FrAug improves the performances of Autoformer by 28% in ETTm1 with a predicted length of 3. In exchange rate with a predicted length of 3, it improves the performance of Autoformer by 50% and improves the performance of Informer by 79%. However, no significant performance boost is observed in DLinear. We find that the model capacity of DLinear is extremely low in short-term forecasting tasks. For example, when the horizon is 3, the number of parameters in DLinear is 6x look-back window size, while other models have more than millions of parameters. Such low model capacity makes DLinear hard to benefit from augmented data.

## A.5 THE OVER-FITTING PROBLEM

Generally speaking, a large gap between the training loss and the test loss, the so-called generalization gap, is an indicator of over-fitting.

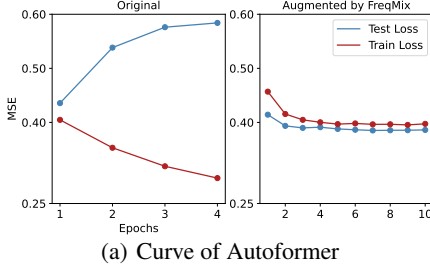 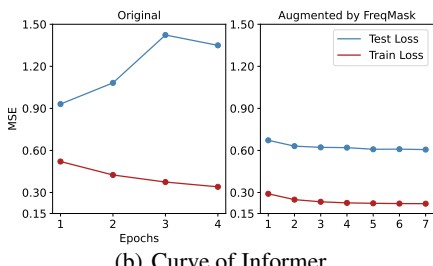

(a) Curve of Autoformer        (b) Curve of Informer

Figure 5: The over-fitting problem of Autoformer(a) and Informer(b). We plot the training(red) and testing(blue) curve in the ETTh1 dataset and predict length 96. The X axis is the epoch and the Y axis is the loss. Models are trained with the early stop policy. The testing curve is much more flattened after applying FreqMask and FreqMix.

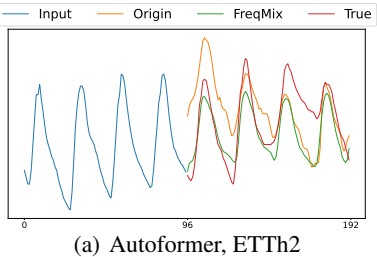 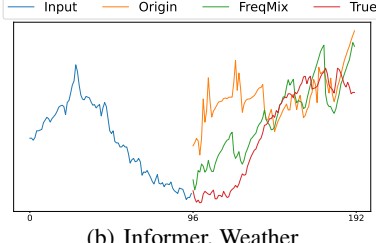

(a) Autoformer, ETTh2        (b) Informer, Weather

Figure 6: Visualization of the forecasting curves of different methods. The X axis is the time steps, and the Y axis shows the forecasting value. With FrAug, models are less likely to overfit.

**For long-term forecasting** (Sec 4.2 of the main paper ), Figure 5 demonstrates training loss and test error curves from deep models Autoformer and Informer. Without FrAug, the training loss of Autoformer and Informer decease with more training epochs, but the test errors increase. In contrast, when FrAug is applied to include more training samples, the test loss can decrease steadily until it is stable. This result clearly shows the benefits of FrAug.

In Figure 6, we visualize some prediction results with/without FrAug. Without FrAug, the model can hardly capture the scale of data, and the predictions show little correlation to the look-back window, i.e, there is a large gap between the last value of the look-back window and the first value of the predicted horizon. This indicates that the models are over-fitting. In contrast, with FrAug, the prediction is much more reasonable.

**In short-term forecasting**, FrAug also mitigates the overfitting problem. We present some visualizations of the training and testing curve in Fig. 7. FrAug can effectively reduce the generalization gap.

## A.6    More results of cold-start forecasting

We simulate the cold-start forecasting tasks by reducing the training samples of each dataset to the last 1%. For example, when the look-back window and horizon are both 96, the 8640 data points in the training set of ETTh1 can form 8448(8640 - 96 - 96) training samples. We use the last 1% of training samples (the 8364th-8448th) for model training. In total, we only use 279(84 + 96 + 96) continuous data points. This is similar to the situation where we train a model to predict the sale curve of a new product based on just a few days' sale data.

In the main paper, we only present part of the results of FrAug in cold-start forecasting. Here we present all the results in Table 7. We can observe that FrAug consistently improves the performances of the model in cold-start forecasting by a large margin. In some datasets, i.e, exchange rate, models trained with FrAug can achieve comparable to those trained on the full dataset. Surprisingly, the performances of the models are sometimes better than those trained on the full dataset, i.e, Informer in Exchange rate and ETTh2. This indicates that FrAug is an effective tool to enlarge the dataset in cold-start forecasting.

Table 6: Performance of models in short-term forecasting tasks. Performances are measured by MSE. The **best results** are highlighted in **bold**.

| Dataset | Pred | DLinear | | | FEDformer | | | Autoformer | | | Informer | | |
|---|---|---|---|---|---|---|---|---|---|---|---|---|---|
| | | Origin | FreqMask | FreqMix | Origin | FreqMask | FreqMix | Origin | FreqMask | FreqMix | Origin | FreqMask | FreqMix |
| ETTh1 | 3 | **0.168** | 0.170 | 0.177 | 0.200 | **0.192** | 0.198 | 0.260 | 0.241 | **0.209** | 0.264 | **0.191** | 0.219 |
| | 6 | 0.241 | **0.233** | **0.233** | 0.250 | 0.244 | **0.242** | 0.339 | 0.329 | **0.305** | 0.482 | **0.311** | 0.322 |
| | 12 | 0.307 | **0.286** | **0.286** | 0.293 | 0.289 | **0.288** | 0.380 | 0.371 | **0.331** | 0.649 | **0.375** | 0.377 |
| | 24 | 0.319 | **0.317** | 0.318 | 0.312 | **0.305** | **0.305** | **0.374** | 0.377 | 0.406 | 0.690 | **0.406** | 0.443 |
| ETTh2 | 3 | **0.083** | 0.084 | 0.085 | 0.151 | 0.139 | **0.136** | 0.171 | 0.146 | **0.140** | 0.288 | 0.163 | **0.141** |
| | 6 | 0.104 | **0.103** | **0.103** | 0.166 | 0.155 | **0.152** | 0.231 | 0.203 | **0.177** | 0.577 | **0.266** | 0.292 |
| | 12 | 0.131 | **0.130** | **0.130** | 0.187 | **0.176** | 0.177 | 0.247 | 0.222 | **0.202** | 1.078 | **0.370** | 0.497 |
| | 24 | 0.168 | **0.166** | 0.167 | 0.216 | 0.205 | **0.204** | 0.298 | 0.249 | **0.246** | 1.218 | **0.512** | 1.127 |
| ETTm1 | 3 | **0.062** | 0.063 | 0.064 | 0.093 | **0.089** | 0.090 | 0.255 | 0.194 | **0.184** | 0.091 | 0.074 | **0.072** |
| | 6 | **0.088** | **0.088** | 0.090 | 0.120 | **0.114** | **0.114** | 0.270 | 0.188 | **0.186** | 0.130 | **0.108** | 0.114 |
| | 12 | 0.138 | **0.136** | 0.137 | 0.171 | **0.168** | **0.168** | 0.291 | **0.253** | 0.262 | 0.251 | **0.175** | 0.192 |
| | 24 | 0.211 | **0.209** | **0.209** | **0.279** | **0.279** | 0.281 | 0.418 | **0.335** | 0.346 | 0.320 | **0.277** | 0.362 |
| ETTm2 | 3 | **0.044** | **0.044** | **0.044** | 0.068 | **0.060** | 0.061 | 0.095 | 0.087 | **0.076** | 0.071 | **0.055** | **0.055** |
| | 6 | **0.056** | **0.056** | **0.056** | 0.080 | **0.074** | **0.074** | 0.124 | **0.108** | 0.110 | 0.100 | **0.077** | 0.086 |
| | 12 | **0.074** | **0.074** | **0.074** | 0.096 | 0.092 | **0.091** | 0.124 | **0.113** | **0.113** | 0.142 | **0.104** | 0.124 |
| | 24 | **0.098** | **0.098** | 0.100 | 0.115 | **0.112** | **0.112** | 0.152 | **0.131** | 0.138 | 0.235 | **0.163** | 0.202 |
| Exchange | 3 | **0.005** | **0.005** | **0.005** | 0.031 | **0.026** | 0.028 | 0.039 | 0.024 | **0.018** | 0.422 | **0.088** | 0.297 |
| | 6 | **0.008** | 0.009 | **0.008** | 0.035 | **0.030** | **0.030** | 0.031 | **0.021** | 0.023 | 0.575 | **0.125** | 0.442 |
| | 12 | **0.014** | **0.014** | **0.014** | 0.044 | **0.039** | 0.040 | 0.054 | **0.028** | 0.030 | 0.581 | **0.132** | 0.470 |
| | 24 | **0.024** | **0.024** | **0.024** | 0.054 | 0.050 | **0.049** | 0.061 | **0.043** | 0.049 | 0.583 | **0.255** | 0.562 |
| Electricity | 3 | **0.070** | 0.081 | 0.087 | 0.142 | **0.129** | 0.130 | 0.147 | **0.128** | 0.129 | 0.233 | **0.182** | 0.187 |
| | 6 | **0.085** | 0.090 | 0.092 | 0.149 | **0.137** | **0.137** | 0.152 | **0.135** | 0.136 | 0.271 | 0.211 | **0.210** |
| | 12 | **0.099** | 0.106 | 0.110 | 0.157 | 0.146 | **0.145** | 0.158 | **0.140** | 0.141 | 0.286 | **0.217** | 0.222 |
| | 24 | **0.110** | **0.110** | 0.111 | 0.164 | **0.153** | **0.153** | 0.176 | **0.139** | 0.141 | 0.292 | **0.229** | 0.231 |
| Traffic | 3 | **0.308** | 0.326 | 0.341 | 0.537 | **0.506** | 0.510 | 0.563 | **0.515** | 0.519 | 0.597 | **0.580** | 0.586 |
| | 6 | **0.342** | 0.347 | 0.352 | 0.548 | **0.519** | **0.519** | 0.564 | **0.526** | 0.527 | 0.619 | 0.609 | **0.607** |
| | 12 | **0.360** | 0.372 | 0.380 | 0.547 | 0.525 | **0.521** | 0.565 | **0.530** | 0.532 | 0.634 | **0.608** | 0.625 |
| | 24 | **0.371** | **0.371** | 0.372 | 0.548 | 0.532 | **0.527** | 0.560 | 0.534 | **0.527** | 0.672 | 0.632 | **0.622** |
| Weather | 3 | 0.049 | **0.047** | **0.047** | 0.086 | **0.080** | 0.081 | 0.156 | **0.115** | 0.131 | 0.068 | 0.058 | **0.056** |
| | 6 | **0.061** | **0.061** | **0.061** | 0.099 | 0.093 | **0.090** | 0.158 | **0.123** | 0.128 | 0.074 | **0.066** | 0.067 |
| | 12 | **0.079** | **0.079** | **0.079** | 0.138 | **0.116** | 0.122 | 0.172 | **0.139** | 0.142 | 0.111 | **0.086** | 0.094 |
| | 24 | **0.104** | 0.105 | 0.105 | 0.153 | 0.151 | **0.140** | 0.182 | **0.147** | 0.158 | 0.189 | **0.126** | 0.161 |

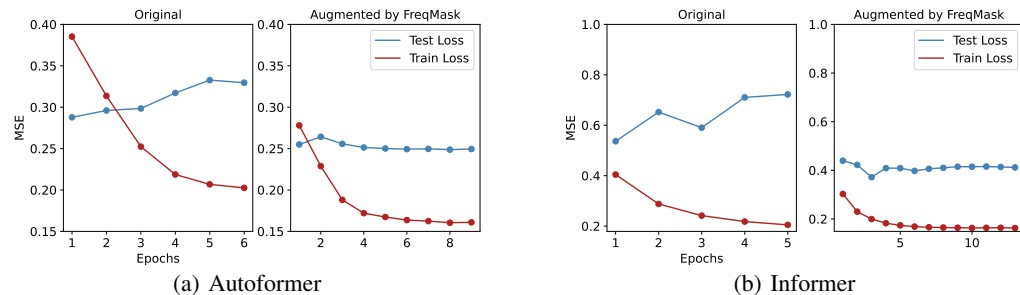

(a) Autoformer       (b) Informer

Figure 7: The over-fitting problem of recent SOTA models in short-term forecasting tasks. We plot the training and testing curve of Autoformer in ETTh2 dataset with predict length 24, and of Informer in ETTh1 with predict length 12. The testing curve are much more flatten after applying FreqMask and FreqMix.

## A.7 MORE RESULTS OF TEST-TIME TRAINING

We present the average test loss of 20 parts of the dataset in Table 8. FrAug improves the performance of the models in most cases. We also observe that FreqMask generally achieves better performance than FreqMix. We attribute this to the fact that FreqMix augments a series by mixing its frequency components with another series. For data with new distribution, mixing with historical data might weaken its ability to help the model fit into the new distribution.

## A.8 DISCUSSIONS

**Preserving dominant component:** One variant of FreqMask is preserving the dominant ones when masking frequency components. The following Table 9 shows the best results on Etth1 of FreqMask with/ without retaining the 10 frequency components with the largest amplitudes. As can be observed,

Table 7: Performance of models trained with the last 1% training samples compared with those trained with the full training set. Performances are measured by MSE. The **best results** are highlighted in **bold** (row full data is not included).

| Dataset | Pred | DLinear | | | | FEDformer | | | | Autoformer | | | | Informer | | | |
|---|---|---|---|---|---|---|---|---|---|---|---|---|---|---|---|---|---|
| | | 1% Data | FreqMask | FreqMix | Full Data | 1% Data | FreqMask | FreqMix | Full Data | 1% Data | FreqMask | FreqMix | Full Data | 1% Data | FreqMask | FreqMix | Full Data |
| ETTh1 | 96 | 0.468 | **0.467** | 0.484 | 0.381 | 0.636 | **0.457** | 0.469 | 0.374 | 0.693 | **0.485** | 0.512 | 0.449 | 1.542 | **0.767** | 1.518 | 0.931 |
| | 192 | 0.666 | **0.546** | 0.549 | 0.405 | 0.659 | **0.549** | 0.617 | 0.425 | 0.753 | 0.608 | **0.594** | 0.463 | 1.508 | **1.131** | 1.518 | 1.010 |
| | 336 | 0.901 | **0.594** | 0.617 | 0.439 | 0.665 | **0.589** | 0.662 | 0.456 | 0.661 | 0.583 | **0.580** | 0.495 | 1.509 | **1.042** | 1.579 | 1.036 |
| | 720 | 0.761 | **0.731** | 0.742 | 0.514 | 0.648 | **0.605** | 0.605 | 0.485 | 0.709 | 0.589 | **0.587** | 0.535 | 1.432 | **1.200** | 1.514 | 1.159 |
| ETTh2 | 96 | 0.572 | **0.351** | 0.464 | 0.295 | 0.398 | **0.391** | 0.394 | 0.339 | 0.490 | **0.409** | 0.414 | 0.432 | 3.115 | **2.276** | 3.002 | 2.843 |
| | 192 | 0.704 | **0.467** | 0.531 | 0.378 | **0.453** | 0.466 | 0.470 | 0.430 | 0.546 | **0.496** | 0.502 | 0.430 | 2.882 | **1.969** | 2.775 | 6.236 |
| | 336 | 0.628 | **0.541** | 0.590 | 0.421 | **0.472** | 0.479 | 0.483 | 0.519 | 0.498 | **0.476** | 0.484 | 0.482 | 3.082 | **1.815** | 2.709 | 5.418 |
| | 720 | 0.662 | **0.640** | 1.005 | 0.696 | 0.457 | **0.455** | 0.456 | 0.474 | 0.479 | **0.471** | 0.471 | 0.471 | 3.017 | **1.814** | 2.953 | 3.962 |
| ETTm1 | 96 | 0.392 | **0.371** | 0.374 | 0.300 | 0.743 | **0.546** | 0.616 | 0.364 | 0.692 | **0.630** | 0.656 | 0.552 | 1.652 | **0.692** | 1.943 | 0.626 |
| | 192 | 0.407 | **0.388** | 0.389 | 0.335 | 0.745 | **0.533** | 0.541 | 0.406 | 0.665 | **0.649** | 0.652 | 0.559 | 1.653 | **0.755** | 1.904 | 0.730 |
| | 336 | 0.432 | **0.416** | 0.418 | 0.368 | 0.750 | **0.616** | 0.709 | 0.446 | **0.625** | 0.632 | 0.626 | 0.605 | 1.720 | **0.801** | 1.812 | 1.037 |
| | 720 | 0.490 | **0.471** | 0.472 | 0.425 | 0.743 | **0.660** | 0.710 | 0.533 | 0.713 | **0.697** | 0.699 | 0.755 | 1.914 | **0.919** | 1.979 | 0.972 |
| ETTm2 | 96 | 0.396 | **0.202** | 0.317 | 0.171 | 0.293 | **0.255** | 0.275 | 0.189 | 0.294 | **0.258** | 0.289 | 0.288 | 2.548 | **2.206** | 2.640 | 0.389 |
| | 192 | 0.795 | **0.254** | 0.650 | 0.235 | 0.342 | **0.330** | 0.333 | 0.253 | 0.395 | **0.355** | 0.426 | 0.274 | 2.703 | **2.287** | 2.675 | 0.813 |
| | 336 | 0.412 | **0.381** | 0.420 | 0.305 | 0.400 | **0.389** | 0.393 | 0.327 | 0.397 | 0.399 | 0.407 | 0.335 | 3.633 | **2.382** | 3.055 | 1.429 |
| | 720 | 0.837 | **0.721** | 0.929 | 0.412 | 0.477 | **0.474** | 0.481 | 0.438 | **0.481** | 0.529 | 0.562 | 0.437 | 2.812 | **2.074** | 2.529 | 3.863 |
| Exchange | 96 | 0.280 | 0.174 | **0.139** | 0.079 | **0.160** | 0.166 | 0.166 | 0.135 | 0.247 | **0.166** | 0.208 | 0.145 | 1.674 | **0.384** | 1.220 | 0.879 |
| | 192 | 0.549 | 0.258 | **0.252** | 0.205 | **0.266** | 0.278 | 0.288 | 0.271 | 0.487 | **0.304** | 0.369 | 0.385 | 1.651 | **0.420** | 1.313 | 1.147 |
| | 336 | 1.711 | **0.777** | 1.205 | 0.309 | **0.430** | 0.520 | 0.523 | 0.454 | **0.579** | 0.791 | 0.832 | 0.453 | 1.849 | **0.798** | 1.330 | 1.562 |
| | 720 | 0.897 | **0.820** | 0.897 | 1.029 | **0.927** | 0.942 | 0.942 | 1.140 | 1.114 | **1.085** | 1.091 | 1.087 | 1.827 | **1.027** | 1.597 | 2.919 |
| Electricity | 96 | 0.196 | **0.172** | 0.176 | 0.140 | 0.537 | 0.314 | **0.302** | 0.188 | 0.462 | **0.326** | 0.347 | 0.203 | 1.238 | **0.744** | 1.118 | 0.305 |
| | 192 | 0.205 | **0.183** | 0.185 | 0.154 | 0.530 | 0.321 | **0.310** | 0.196 | 0.488 | 0.331 | **0.326** | 0.231 | 1.230 | **0.681** | 0.989 | 0.349 |
| | 336 | 0.218 | **0.197** | 0.200 | 0.169 | 0.533 | 0.337 | **0.335** | 0.212 | 0.518 | 0.410 | **0.381** | 0.247 | 1.216 | **0.767** | 1.025 | 0.349 |
| | 720 | 0.280 | **0.257** | 0.257 | 0.204 | 0.563 | **0.500** | 0.500 | 0.250 | 0.541 | 0.483 | **0.479** | 0.276 | 1.219 | **0.832** | 1.070 | 0.391 |
| Traffic | 96 | 0.764 | **0.466** | 0.486 | 0.410 | 1.360 | 0.791 | **0.787** | 0.574 | 1.238 | 0.761 | **0.746** | 0.624 | 1.613 | **1.066** | 1.262 | 0.736 |
| | 192 | 0.658 | **0.484** | 0.510 | 0.423 | 1.365 | 0.788 | **0.742** | 0.611 | 1.366 | **0.832** | 0.844 | 0.619 | 1.615 | **1.028** | 1.136 | 0.770 |
| | 336 | 0.825 | **0.509** | 0.515 | 0.436 | 1.376 | 0.794 | **0.740** | 0.623 | 1.299 | **0.706** | 0.715 | 0.604 | 1.624 | **1.306** | 1.389 | 0.861 |
| | 720 | 0.908 | **0.539** | 0.578 | 0.466 | 1.400 | 0.904 | **0.761** | 0.631 | 1.325 | **0.786** | 0.795 | 0.703 | 1.638 | **1.475** | 1.541 | 0.995 |
| Weather | 96 | 0.245 | **0.212** | 0.214 | 0.175 | 0.295 | **0.273** | 0.280 | 0.250 | **0.293** | 0.299 | 0.318 | 0.271 | 1.735 | **0.671** | 1.601 | 0.452 |
| | 192 | 0.264 | 0.241 | **0.240** | 0.217 | 0.318 | **0.307** | 0.319 | 0.266 | 0.361 | **0.310** | 0.341 | 0.315 | 1.950 | **0.517** | 1.874 | 0.466 |
| | 336 | 0.294 | 0.284 | **0.283** | 0.265 | **0.367** | 0.372 | 0.390 | 0.368 | **0.384** | 0.400 | 0.431 | 0.345 | 1.608 | **0.536** | 1.546 | 0.499 |
| | 720 | 0.374 | **0.366** | 0.372 | 0.324 | 0.428 | **0.419** | 0.426 | 0.397 | **0.458** | 0.472 | 0.485 | 0.452 | 1.234 | **0.642** | 1.256 | 1.260 |

Table 8: Average test loss of test time training experiment. Performances are measured by MSE. The **best results** are highlighted in **bold**.

| Model | Method | ETTh1 | | | | ETTh2 | | | | ILI | | | |
|---|---|---|---|---|---|---|---|---|---|---|---|---|---|
| | | 96 | 192 | 336 | 720 | 96 | 192 | 336 | 720 | 24 | 36 | 48 | 60 |
| Dlinear | Origin | 0.473 | 0.647 | 1.365 | 1.317 | 0.382 | 0.604 | 0.825 | 0.938 | 0.831 | 0.713 | 0.723 | 0.842 |
| | FreqMask | 0.428 | **0.522** | **0.599** | **0.775** | 0.356 | **0.448** | **0.540** | **0.710** | 0.861 | **0.663** | **0.652** | 0.954 |
| | FreqMix | **0.427** | 0.523 | 0.874 | 1.216 | **0.353** | 0.516 | 0.687 | 0.807 | **0.809** | 0.731 | 0.686 | **0.743** |
| FEDformer | Origin | 6.223 | 10.330 | 1.600 | 1.888 | 25.937 | 3.290 | 1.466 | 2.445 | 1.460 | 1.240 | 1.121 | 0.977 |
| | FreqMask | 5.412 | 2.791 | **0.944** | 1.523 | 0.554 | 2.993 | **0.592** | 1.107 | 0.926 | **0.860** | **0.768** | **0.733** |
| | FreqMix | **1.334** | **1.222** | 1.827 | **1.328** | 0.982 | **0.881** | 1.724 | **1.054** | 1.075 | 1.206 | 0.837 | 0.777 |
| Autoformer | Origin | 1.457 | 1.403 | 1.216 | 2.175 | 34.503 | 2.633 | 1.311 | 1.106 | 1.528 | 1.471 | 0.905 | 1.027 |
| | FreqMask | 0.786 | **0.845** | **0.867** | **1.116** | 0.506 | **0.556** | **0.599** | 0.848 | 0.965 | **0.844** | **0.749** | **0.735** |
| | FreqMix | **0.725** | 0.962 | 1.039 | 1.658 | 0.588 | 0.677 | 0.731 | 0.959 | 1.136 | 1.230 | 0.781 | 0.746 |
| Informer | Origin | 2.199 | 2.274 | 2.190 | 2.726 | 1.755 | 1.797 | 1.738 | 2.109 | 1.989 | 1.836 | 2.083 | 1.613 |
| | FreqMask | **1.052** | **0.999** | **1.009** | **1.034** | **1.004** | **1.045** | **1.006** | **1.111** | **0.908** | **0.947** | **0.749** | 0.807 |
| | FreqMix | 2.147 | 2.102 | 2.113 | 2.676 | 1.610 | 1.743 | 1.717 | 2.116 | 2.403 | 1.963 | 1.915 | 1.662 |
| Film | Origin | 0.787 | 1.325 | 1.238 | 1.817 | 0.785 | 0.749 | 0.808 | 1.238 | 1.830 | 1.485 | 0.939 | 1.059 |
| | FreqMask | **0.677** | **0.677** | **0.942** | **0.942** | 0.473 | 0.473 | **0.601** | **0.833** | **0.977** | **0.822** | **0.734** | **0.741** |
| | FreqMix | 0.684 | 0.684 | 1.109 | 1.109 | **0.443** | **0.443** | 0.756 | 1.217 | 1.141 | 1.322 | 0.757 | 0.789 |

the original FreqMask achieves better results in most cases compared to keeping the dominant frequency components in masking. We argue that, on the one hand, as the original data are kept during training, the dominant frequency components do exist in many training samples. On the other hand, masking the dominant frequency is useful for the forecaster to learn the temporal relations of other frequency components.

**Combining FreqMask and FreqMix:** FreqMask and FreqMix do not conflict with each other. Therefore, it is possible to combine them for data augmentation. Table 10 shows the results of such combinations in long-term forecasting. However, we cannot observe many improvements. We hypothesize that either FreqMask or FreqMix has already alleviated the overfitting issue of the original model with the augmented samples.

**Important Factors influencing the performance of FrAug:** We find that whether the data is sufficient for model training is an important factor influencing the performance of FrAug. FrAug cannot significantly improve the model's performance in situations where the dataset is large, or the model has a low capacity. For example, in the experiment of the full Electricity dataset and DLinear

Table 9: Results for keeping 10 frequency components with the largest amplitude on ETTh1 Dataset.

| Method | Horizon | DLinear | FEDformer | Film | Autoformer | LightTS |
|--------|---------|---------|-----------|------|------------|---------|
| Keep Dominant | 96 | **0.369** | 0.373 | 0.528 | 0.496 | **0.419** |
| | 192 | 0.414 | 0.428 | 0.754 | 0.751 | 0.434 |
| | 336 | **0.428** | 0.446 | 0.713 | 0.688 | 0.578 |
| | 720 | 0.492 | 0.489 | 0.991 | 0.985 | 0.621 |
| FrAug | 96 | 0.379 | **0.372** | **0.446** | **0.419** | **0.419** |
| | 192 | **0.403** | **0.417** | **0.461** | **0.426** | **0.427** |
| | 336 | 0.435 | 0.457 | **0.490** | **0.476** | **0.572** |
| | 720 | **0.472** | **0.474** | **0.483** | **0.501** | **0.617** |

Table 10: Performance of combination of FreqMask and FreqMix in ETTh2. Performances are measured by MSE. The **best results** are highlighted in **bold**.

| Model | | DLinear | | | | Autoformer | | |
|-------|--------|----------|---------|---------|--------|------------|---------|---------|
| Method | Origin | FreqMask | FreqMix | Combine | Origin | FreqMask | FreqMix | Combine |
| 96 | 0.295 | **0.277** | **0.277** | **0.277** | 0.432 | **0.346** | 0.351 | 0.349 |
| 192 | 0.378 | 0.338 | 0.342 | **0.334** | 0.430 | 0.422 | 0.423 | **0.419** |
| 336 | **0.421** | 0.432 | 0.449 | 0.428 | 0.482 | **0.447** | 0.455 | 0.458 |
| 720 | 0.696 | 0.588 | 0.636 | **0.546** | 0.471 | 0.462 | **0.449** | 0.455 |

model (which has only one linear layer), the improvement brought by FrAug is subtle. In contrast, in the experiment of the last 1% Electricity dataset and DLinear model or the experiment of the full Electricity dataset and Autoformer model, FrAug brings a large improvement. Therefore, FrAug is especially useful for applications with data scarcity, such as cold-start problems and test-time training.

