# OpenReview forum: "FrAug: Frequency Domain Augmentation for Time Series Forecasting"
_ICLR.cc/2024/Conference — Submitted to ICLR 2024_

### Official Review · Reviewer_bZGH · 2023-10-15

**Soundness:** 3 good
**Presentation:** 2 fair
**Contribution:** 1 poor
**Rating:** 5
**Confidence:** 4

**Summary:**

This paper proposes frequency-based time-series augmentation, FrAug, for long-term forecasting task. The first augmentation method is frequency masking and the second one is frequency mixing, where Fourier transform is used to get the amplitude of frequency. In scenarios where time-series data is scarce, FrAug shows SOTA performance and also improves the performance of most of time series forecasting model comparing to other augmentation methods.

**Strengths:**

1. Very clear motivation in FrAug, supported by practical applications.
2. Simple and effective augmentation method.
3. Realistic experiment setting and superiority when a time series itself is scarce, including the situations such as cold start and distribution shift.

**Weaknesses:**

1. Technical novelty is a bit short. Frequency masking is already used by SpecAugment[1] in automatic speech recognition and mixing strategy is common in computer vision such as MixUp.

2. There is no theory or heuristic about how to mask or mix frequencies. For example, how to select masked frequency ratio and which frequency should be masked first? When mixing frequencies, are you averaging two frequency amplitudes without any weight? Is there any chance that randomly masked frequency would be crucial? This seems the main hole in this paper.

3. Lack of evidence in claims.
(1) Aggressive augmentation methods like cropping or warping can introduce missing values or changes in periodicity, which are not suitable for TSF models.
(2) Data from other sources are often not useful (even if they are the same kinds of data), as the underlining dynamic systems that generate the time series data could be vastly different.
These statements is not proved and does not have enough reference. (1) is partly proved by a single dataset experiment but does not support the claim sufficiently.

**Questions:**

0. Please see weaknesses for the questions.
1. The second paragraph of Introduction seems to have duplicated sentences and miss a blank. How about refining the paragraph?

---

### Official Review · Reviewer_SMpB · 2023-11-01

**Soundness:** 3 good
**Presentation:** 2 fair
**Contribution:** 2 fair
**Rating:** 5
**Confidence:** 4

**Summary:**

This paper presents a pair of techniques in data augmentation of time series forecasting. Different from conventional perturbations in time domain, it proposes to manipulate the frequency spectrum of given time series by masking/mixing frequency modes, so that the changes will be applied to both history and future in a consistent way. Experiments that add proposed DA methods to SOTA TSF models support their effectiveness.

**Strengths:**

1. The paper is overall well-written and easy to follow, with the core idea highlighted throughout the full text.
2. The motivation behind the frequency domain DA is very reasonable.
3. The empirical results are quite strong. In particular, the effectiveness on dealing with distribution shift and cold start are quite impressive and promising.

**Weaknesses:**

1. The claim of being "the first work that systematically investigates data augmentation techniques for the TSF task" is exaggerated. There have been earlier works that discuss DA techniques, including frequency domain techniques, such as [1]. Beyond TSF task, the proposed method is also not novel in similar domains, such as frequency mode masking in ASR [2]. Such a paper is also not sufficient for a "systematical investigation".
2. It's interesting that FrAug does not improve in any cases with `electricity` dataset. It might be necessary to look into it.



[1] Wen, Q., Sun, L., Yang, F., Song, X., Gao, J., Wang, X. and Xu, H., 2020. Time series data augmentation for deep learning: A survey. arXiv preprint arXiv:2002.12478.
[2] Park, D.S., Chan, W., Zhang, Y., Chiu, C.C., Zoph, B., Cubuk, E.D. and Le, Q.V., 2019. Specaugment: A simple data augmentation method for automatic speech recognition. arXiv preprint arXiv:1904.08779.

**Questions:**

1. For multivariate time series, how do you manipulate frequency maps of different features? If processed independently, how do you account for the inter-feature correlations?
2. Would it be a even simpler and more effective way to add noise in the frequency domain?

---

### Official Review · Reviewer_j37M · 2023-11-01

**Soundness:** 3 good
**Presentation:** 3 good
**Contribution:** 1 poor
**Rating:** 3
**Confidence:** 4

**Summary:**

This paper introduces the problem of data augmentation for time series forecasting (TSF) tasks.  In TSF where accurate prediction results are required, existing temporal enhancement methods destroy fine-grained temporal relationships within time series segments, resulting in poor prediction accuracy. To solve this problem, this paper proposes a frequency domain augmentation technique called FrAug, which can maintain the semantic consistency of augmented data label pairs in prediction. The authors conduct extensive experiments on eight benchmark datasets and show that FrAug can improve the prediction accuracy of TSF models even when the training dataset is small. They also demonstrate that applying FrAug during training at test time can significantly improve forecast accuracy for time series with significant distribution changes. The main contributions of this paper include a systematic study of TSF data augmentation techniques, the proposal of FrAug, and the demonstration of its effectiveness in improving prediction performance.

**Strengths:**

+ The paper is well-written and well-organized.
+ The paper proposes frequency domain augmentation techniques, named FrAug, to preserve the semantic consistency of augmented data-label pairs in forecasting.
+ FrAug improves the forecasting accuracy of TSF models, even with a small training dataset.
+ FrAug can be applied during test-time training to mitigate distribution shift problems and improve forecasting accuracy.
+ The paper provides extensive experimental results on eight benchmark datasets to support the effectiveness.

**Weaknesses:**

- To the best of my knowledge, the method is not novel. The frequency-based data augmentation was introduced in existing work like [1], which is not included in literature reviews.
- There should be some hyper-parameters in both FreqMix and FreqMask, e.g. the range of frequency to be masked, which are not elaborated. Also, a sensitiveness analysis should be conducted to investigate such hyper-parameters.

[1] Supervised Contrastive Few-Shot Learning for High-Frequency Time Series, AAAI-2023

**Questions:**

1. Elaborate on the superioty of the proposed method beyond [1].

2. Provide the selection schemes of potential hyper-parameters in FreqMix and FreqMask. Provide the sensitiveness analysis if possible.

---

### Official Review · Reviewer_Arja · 2023-11-03

**Soundness:** 2 fair
**Presentation:** 2 fair
**Contribution:** 2 fair
**Rating:** 5
**Confidence:** 4

**Summary:**

This paper addresses the challenge of data augmentation in the context of time series forecasting (TSF), where maintaining fine-grained temporal relationships is crucial for accurate predictions. Existing data augmentation techniques developed for time series classification and anomaly detection may disrupt these relationships, leading to reduced forecasting accuracy. To overcome this limitation, the paper introduces FrAug, a set of frequency domain augmentation techniques that ensure the semantic consistency of data-label pairs for forecasting tasks. The study includes extensive experiments on eight benchmark datasets, employing state-of-the-art TSF deep models which demonstrate that FrAug enhances forecasting accuracy.

**Strengths:**

1. this writing of this paper is easy to understand.
2. this paper considers the frequency domain augmentation, which is somehow novel to time series forecasting.

**Weaknesses:**

To me, my biggest concern is about the experimental results.
1. The experiments are conducted and compared with Informer, Autoformer, MICN, etc, but they don't compare performances with the more recent well-known methods, such as PatchTST [1].
2. Why the authors compare with transformer-based models in short-term settings, because these models are focusing on long-term forecasting.
3. there is no specific analysis towards the augmenetations, such as case studies, parameter analysis or deep visualizations.

[1] A Time Series is Worth 64 Words: Long-term Forecasting with Transformers.

**Questions:**

See weakness

---

### Official Review · Reviewer_Wo4Z · 2023-11-05

**Soundness:** 3 good
**Presentation:** 2 fair
**Contribution:** 1 poor
**Rating:** 5
**Confidence:** 4

**Summary:**

The paper applies data augmentations in the frequency domain to improve the performance of time series forecasting models. Specifically, they apply two augmentations: 1) frequency masking: where some frequencies are dropped, and 2) frequency mixing: where some frequencies are mixed. To ensure consistency, they apply the transformations to the input and the forecasting window. After applying the augmentations in some datasets for training a group of standard forecasting models, the authors demonstrate that the method improves the performance, compared to other data augmentation techniques. Moreover, they run experiments to measure the performance of the techniques under "cold start" setups, i.e. when little training data is available.

**Strengths:**

- The paper is well written and understandable in most of the parts.
- The method is methodologycal correct and makes sense.
- The authors motivate the importance of the problem.

**Weaknesses:**

- My biggest concern is that the contribution of this work is very limited. Augmentations in the frequency domain for time series have been discussed before [1][2][3], but the authors did not mention any previous work on this. Thus claiming to be the "first" ones in this seems like an overstatement. In fact, the proposed data augmentations are very similar to previous work [3] such as the masking of frequencies.
- The experiment setup is sound but also limited. The authors only run experiments in 8 datasets, while time series forecasting literature considers way more. For instance, M3 and M4 datasets are not present. Moreover, they only compare to 4 Data augmentation baselines.
- The improvement is marginal. The authors do not specify whether the results correspond to the average of different runs. If that is the case, they should provide standard deviations to evaluate the significance of the results.
- A discussion on the time overhead created by the data augmentations is lacking. I would like to see how the training time increases depending on the different data augmentations that are used.
- The data augmentations helps especially in few-shot setups. Thus, a well-regularized model will have no gain from the introduced method.


[1] Wen et al. "Time Series Data Augmentation for Deep Learning: A Survey".

[2] Eyobu et al. "Feature representation and data augmentation for human activity classification based on wearable IMU sensor data using a deep LSTM neural network".

[3] Park et al. "SpecAugment: A simple data augmentation method for automatic speech recognition".

**Questions:**

- Are you using the same hyperparameter value (mask rate, mix rate) for all the experiments or do you tune per time series?
- Why does it look like the method is not working for DLinear compared to the other models?
- Basically, this paper introduces just another hyperparameter when learning forecasting models. Why should this be included in a hyperparameter search space when there are also other regularization methods such as L1 or L2 regularization? Do you have any evidence that the regularization effect induced by this data augmentation is stronger than other regularization methods?

---

### Meta-Review · Area_Chair_pdFN · 2023-12-04

**Metareview:**

The manuscript was reviewed by five reviewers and all unanimously suggested rejection. Moreover, authors did not respond to the reviews. Following the reviews and authors lack of response, I also recommend rejection.

**Justification For Why Not Higher Score:**

All the raised issues by the reviewers are valid and authors chose to not respond.

**Justification For Why Not Lower Score:**

N/A

---

### Decision · Program_Chairs · 2024-01-16

Reject